# Effect of Ni Addition on Catalytic Performance of Fe$_{87}$Si$_5$B$_2$P$_3$Nb$_2$Cu$_1$ Amorphous Alloys for Degrading Methylene Blue Dyes

**Jinfang Shi** 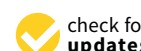**, Bingying Ni, Jingjing Zhang, Chen Wu, Daowen Cheng, Yue Chi, Hongli Wang, Minggang Wang and Zhankui Zhao \***

College of Material Science and Engineering, Key Laboratory of Advanced Structural Materials, Ministry of Education, Changchun University of Technology, Changchun 130012, China; sjf8272@163.com (J.S.); nby6566@163.com (B.N.); zhangjj@ccut.edu.cn (J.Z.); wuchen@ccut.edu.cn (C.W.); chengdaowen@ccut.edu.cn (D.C.); yuechi@ccut.edu.cn (Y.C.); wanghongli@ccut.edu.cn (H.W.); wangminggang@ccut.edu.cn (M.W.)

\* Correspondence: zhaozk@ccut.edu.cn; Tel.: +86-431-8571-6644

**Abstract:** Fe-based amorphous alloys have shown great potential in degrading azo dyes and other organic pollutants. It has been widely investigated as a kind of environmentally friendly material for wastewater remediation. In this paper, we studied the effect of Ni addition on the catalytic performance of Fe$_{87}$Si$_5$B$_2$P$_3$Nb$_2$Cu$_1$ amorphous alloy for degradation of methylene blue dyes and analyzed the reaction mechanism. (Fe$_{87}$Si$_5$B$_2$P$_3$Nb$_2$Cu$_1$)$_{86}$Ni$_{14}$ amorphous powder with desirable performance was produced by specific ball milling durations. Characterization of the Fe$_{87}$Si$_5$B$_2$P$_3$Nb$_2$Cu$_1$ and (Fe$_{87}$Si$_5$B$_2$P$_3$Nb$_2$Cu$_1$)$_{86}$Ni$_{14}$ amorphous alloys prepared by ball milling was performed by XRD and SEM. Fe$_{87}$Si$_5$B$_2$P$_3$Nb$_2$Cu$_1$ and (Fe$_{87}$Si$_5$B$_2$P$_3$Nb$_2$Cu$_1$)$_{86}$Ni$_{14}$ amorphous alloys were used as catalysts to catalyze the degradation of methylene blue dyes, which were detected by UV-VIS near-infrared spectrophotometer. By a series of comparative experiments, it was found that a catalyst dosage of 0.2 g and a reaction temperature of 80 °C were conditions that produced the best catalytic effect. The degradation rate of (Fe$_{87}$Si$_5$B$_2$P$_3$Nb$_2$Cu$_1$)$_{86}$Ni$_{14}$ amorphous alloy to methylene blue dyes prepared by ball milling increased from 67.76% to 99.99% compared with the Fe$_{87}$Si$_5$B$_2$P$_3$Nb$_2$Cu$_1$ amorphous alloy under the same conditions.

**Keywords:** amorphous alloy; catalytic; methylene blue; degradation rate

## 1. Introduction

Industrial wastewater usually contains dye pollutants, which are introduced in the early stages of some production processes. However, the wide use of dyes can cause serious water pollution problems. This is also the main focus of all industrial and manufacturing organizations [1–5]. Methylene blue (MB), a common cationic dye, is widely used in the coloration of cotton, hemp, bamboo, etc. It is one of the most difficult pollutants to remove from water due to its high stability and resistance to decomposition. Excessive use of MB dyes has adverse effects on human health. Therefore, it is important to remove the MB dyes from the sewage. MB is a very effective acrylic dye with a poor effect on natural fibers. However, it is easy for MB to form covalent bonds with suitable polyacrylic acid to form resistant materials. Quite different from crystalline metals, amorphous alloys are solid materials with thermodynamic non-equilibrium characteristics, metastable properties, and disordered atomic arrangement. These peculiarities make metallic glasses excellent catalysts. When the amorphous forming ability is sufficiently large, the properties of amorphous alloys can be adjusted by changing chemical composition. The metastable structure, widely adjustable composition,

and internal and external brittleness of metallic glasses make them excellent materials for degradation of water pollutants.

Fe-based metallic glasses, which have great potential in degrading azo dyes and other organic pollutants, have been widely studied as an environmentally friendly wastewater repairing material [6–8]. Studies have been carried out on the degradation of azo dyes in aqueous solution by various amorphous alloys, such as Fe-based alloys [9–19], Mg-based alloys [20–22], Co-based alloys [23], and Al-based alloys [24], due to their advanced catalytic capability for wastewater remediation. The degradation of azo dyes can be attributed to amorphous or metastable thermodynamic properties of amorphous alloys and the tight surface structure [9,10,16,17]. Amorphous or metastable thermodynamic characteristics make amorphous alloys that process higher energy levels than crystalline alloys. The excited energy state further reduces the activation energy of the degradation reaction and accelerates the degradation rate. On the other hand, the short-range ordered, long-range disordered atomic structure and the incompact surface structures are necessary for dye degradation because the cleavage of –N=N– bonds in dye molecules involves electron transfer from zero-valent iron [25]. In the amorphous alloys used for azo dye degradation, Fe-based amorphous alloys are easy to be recycled and have good degradation efficiency decay due to their superior soft magnetic property and chemical stability. Therefore, investigations of their degradation performance and related mechanisms have attracted much attention from this field. For example, Zhang, L.C. studied $Fe_{73.5}Si_{13.5}B_9Cu_1Nb_3$ amorphous alloy to degrade 3B-A dye [7]. The end result showed almost completed degradation and provided higher development space for catalytic degradation. Zhang, H.F., et al. studied the effect of adding Ni, Co, Cr, and other elements to Fe-Si-B amorphous alloy on the decolorization performance of Acid Orange II solution. The catalytic degradation performance of Co-based amorphous powders prepared by ball milling on Acid Orange II solution was studied, and the degradation rate was over 99% [23,26]. However, it is well known that transition metals have similar special properties, as reported by Zhang, H.F., et al. [26]. The doping of an appropriate amount of transition metal has a beneficial effect on the formation of amorphous alloy. Therefore, whether the addition of other transition metals into amorphous alloy for ball milling will produce similar results remains an open question.

In previous studies, it was proven that the Fe-Si-B amorphous alloy is an excellent Fenton-like catalyst. In this work, we mixed Ni micron powder with $Fe_{87}Si_5B_2P_3Nb_2Cu_1$ amorphous powder in a certain proportion and prepared $(Fe_{87}Si_5B_2P_3Nb_2Cu_1)_{86}Ni_{14}$ amorphous alloy by ball grinding. We compared $Fe_{87}Si_5B_2P_3Nb_2Cu_1$ and $(Fe_{87}Si_5B_2P_3Nb_2Cu_1)_{86}Ni_{14}$ amorphous alloys in the treatment of MB dye's wastewater. The morphology of the powder was observed by SEM, the structure of the powder was analyzed by XRD, and the absorbance of MB in the solution was measured using UV-VIS. We calculated the degradation rate of the reaction and studied the kinetics of the reaction, which proved that the addition of Ni micron powder greatly improved the catalytic performance of iron-based amorphous alloy.

## 2. Experimental

### 2.1. Materials

Fe, Si, B, P, Nb, and Cu with a purity of 99 wt. % were used as raw materials, and the alloy ingot was inductively smelted under the protection of $N_2$ gas, and then the $Fe_{87}Si_5B_2P_3Nb_2Cu_1$ amorphous powder was sprayed by gas atomization. Compared with the $Fe_{73.5}Si_{13.5}B_9Cu_1Nb_3$ amorphous alloy, the amorphous alloys used in this experiment added the P element and increased the content of the Fe element in the preparation process. It gave the alloy good glass forming ability and reduced the cost. High-purity Ni micron powder, MB dyes, and 30 wt. % $H_2O_2$ solutions were purchased from Beijing Chemical Corp. All chemicals were directly used as received without any further purification. Deionized water was used for all experiments.

*2.2. Methods*

2.2.1. Ball milling

Fe$_{87}$Si$_5$B$_2$P$_3$Nb$_2$Cu$_1$ was used as the original material, and a certain proportion of high-purity Ni micron powder was doped for ball milling to prepare (Fe$_{87}$Si$_5$B$_2$P$_3$Nb$_2$Cu$_1$)$_{86}$Ni$_{14}$ amorphous alloy. Firstly, Fe$_{87}$Si$_5$B$_2$P$_3$Nb$_2$Cu$_1$ and high-purity Ni micron powder were separately screened. Secondly, 85 wt. % and 15 wt. % of amorphous powder and Ni micron powder were weighed into a ball mill tank, respectively. This experiment used a 100 mL vacuum carbide ball mill (QM-3SP04, Nanjing University Instrument Factory, Nanjing, China) tank with a ball to material ratio of 10:1. The experiment was ground by wet grinding (wet grinding environment was alcohol), and the rotational speed was controlled at 400 r/min. During the ball milling process, sampling was performed every 5 h for SEM and XRD testing. The best ball milling time of the (Fe$_{87}$Si$_5$B$_2$P$_3$Nb$_2$Cu$_1$)$_{86}$Ni$_{14}$ amorphous alloy was determined by the test results. Finally, (Fe$_{87}$Si$_5$B$_2$P$_3$Nb$_2$Cu$_1$)$_{86}$Ni$_{14}$ amorphous alloy catalyst was produced by ball milling.

2.2.2. Degrading MB dyes

0.1 g of MB was dissolved in 1000 mL of water and stirred thoroughly to prepare 100 mg/L of MB solution. H$_2$O$_2$ solution with an initial concentration of 0.31 mol/L was prepared. We took 50 mL of the 100 mg/L MB solution in a 150 mL beaker for use, then 0.2 g (Fe$_{87}$Si$_5$B$_2$P$_3$Nb$_2$Cu$_1$)$_{86}$Ni$_{14}$ amorphous alloy was weighed into a beaker and stirred with a glass rod to fully mix with the solution. Then, the rotor and 33 mL of H$_2$O$_2$ solution were added to the beaker. The beaker was placed in a water bath of a thermostatic magnetic stirrer for catalytic degradation. The experimental temperatures were 20, 40, 60, and 80 °C, respectively. We took 3 mL of solution with a pipette every 30 min and put it into the centrifugal tube for high-speed centrifugation (8000 r/min centrifugation for 5 min). The supernatant was stored in the sample bottle for UV-VIS analysis (since the MB dyes are easily decomposed by light, they needed to be protected from light by tin foil during the experiment). Since the degradation reaction hardly occurred under the reaction conditions of 20 °C, the results obtained under 20 °C were not shown in this paper.

*2.3. Characterizations*

XRD patterns were carried out by a Bruker D8 Advance X-ray diffractometer (Bruker, Hamburg, Germany) with a Cu Kα X-ray source operating at 9 kV and 200 mA. The surface morphologies before and after ball milling were observed by a Germany Zeiss super high-resolution SUPRA 40 field emission SEM (Carl Zeiss, Jena, Germany). The absorbance of the dye was detected by an Agilent's Cary 5000 UV-VIS Near Infrared Spectrophotometer (Agilent, Walter Bron, Australia).

**3. Results**

*3.1. Subsection*

Bulleted lists

- XRD
- SEM
- EDS
- Degradation rate
- UV-VIS

Figure 1 shows both the XRD patterns of Fe$_{87}$Si$_5$B$_2$P$_3$Nb$_2$Cu$_1$ and Fe$_{87}$Si$_5$B$_2$P$_3$Nb$_2$Cu$_1$ mixed with high purity Ni at different ball milling times to form (Fe$_{87}$Si$_5$B$_2$P$_3$Nb$_2$Cu$_1$)$_{86}$Ni$_{14}$. As can be seen from Figure 1, the particle size of Ni micron powder gradually decreased with the extension of ball milling

time, and the peak width gradually became wider. From Figure 1a, the amorphous diffraction peak of α-Fe is shown at 2θ = 45°, and the crystal plane is (110). The diffraction peak of pure Ni micron powder is shown at 2θ = 44°, 53°, and 77°, corresponding to the crystal planes of (111), (200), and (220), respectively. The XRD diffraction magnified image of amorphous powder and Ni micron powder between 2θ = 40–60° is shown in Figure 1b. From Figure 1, there is still a peak at 2θ = 53°, when the ball milling time did not reach 25 h. This suggests that the Ni micron powders were not completely mixed with the amorphous powders. When the ball milling time exceeded 25 h, we found that the peak height of the amorphous alloy increased, possibly due to the amorphous powder crystallizing or the solid solution of the amorphous powder and Ni micron powder crystallizing. The amorphous powder mixed with Ni micron powder was better when the ball grinding time was 20 h and 25 h. However, when the ball milling time was 25 h, it was found that the diffraction peak height of the pure Ni micron powder decreased. The $(Fe_{87}Si_5B_2P_3Nb_2Cu_1)_{86}Ni_{14}$ amorphous powder had a wider sealing width (FWHM = 0.374), the highest particle size, and more obvious amorphization. The reason why the catalytic activity was higher than 20 h could be attributed to the presence of a small amount of nanocrystals in the amorphous alloy after ball milling for 25 h. As reported by Yao, K., et al. [27], when a small amount of crystallization and a small amount of mixed nanocrystals exist in amorphous alloy, the catalytic performance of amorphous alloy will be improved. Therefore, 25 h was the best ball milling time to achieve the best ball grinding effect.

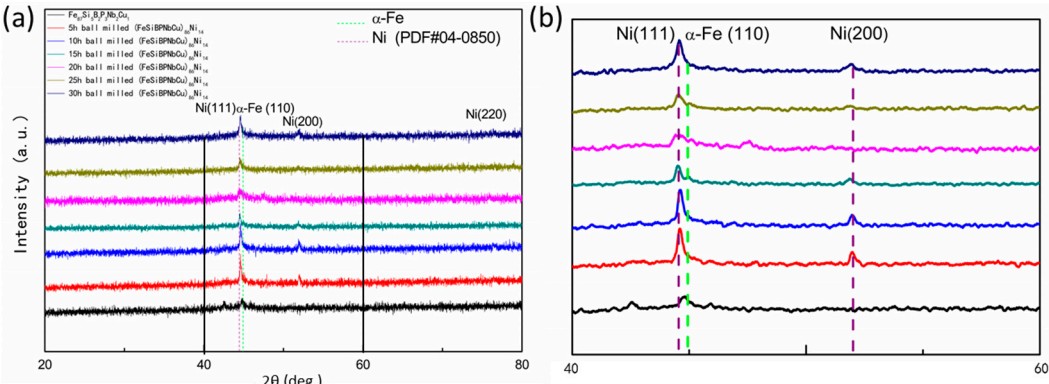

**Figure 1.** (**a**) XRD pattern of $Fe_{87}Si_5B_2P_3Nb_2Cu_1$ amorphous powder and $(Fe_{87}Si_5B_2P_3Nb_2Cu_1)_{86}Ni_{14}$ amorphous powder with different ball milling time (**b**) XRD diffraction enlargement of amorphous alloy between 2θ = 40–60°.

Figure 2 shows the SEM images of the $Fe_{87}Si_5B_2P_3Nb_2Cu_1$ amorphous powder and $(Fe_{87}Si_5B_2P_3Nb_2Cu_1)_{86}Ni_{14}$ amorphous powder. It can be seen from Figure 2a that the $Fe_{87}Si_5B_2P_3Nb_2Cu_1$ amorphous powder had a spherical distribution, smooth surface, uniform particle size, and was kept in a good amorphous state. From Figure 2b, $(Fe_{87}Si_5B_2P_3Nb_2Cu_1)_{86}Ni_{14}$ amorphous powder was distributed in an irregular shape with a rough surface. It can be seen from the illustration in Figure 2b that the size of the particles was about 1 μm, and the particles had good separability. The surface of the powder exhibited high roughness and waviness, which significantly expanded its surface area and provided more active sites for the reaction to improve its catalytic activity. All the samples were scanned, and it was found that the particle size of the amorphous powder decreased with increasing ball milling time. Small powder particles (particle sizes of about 10–20 mm) were either amorphous powder or nickel micron powder. Based on the analysis of SEM and XRD results, the optimal ball milling time was finally determined to be 25 h. As reported by Tang et al. [9,10], the enrichment of metalloid elements in the surface layer is conducive to the formation of an incompact surface layer, which can ensure the effective contact of the inner zero-valent iron atoms with dye solutions, thus keeping the degradation rate of $(Fe_{87}Si_5B_2P_3Nb_2Cu_1)_{86}Ni_{14}$ amorphous powder at a high level.

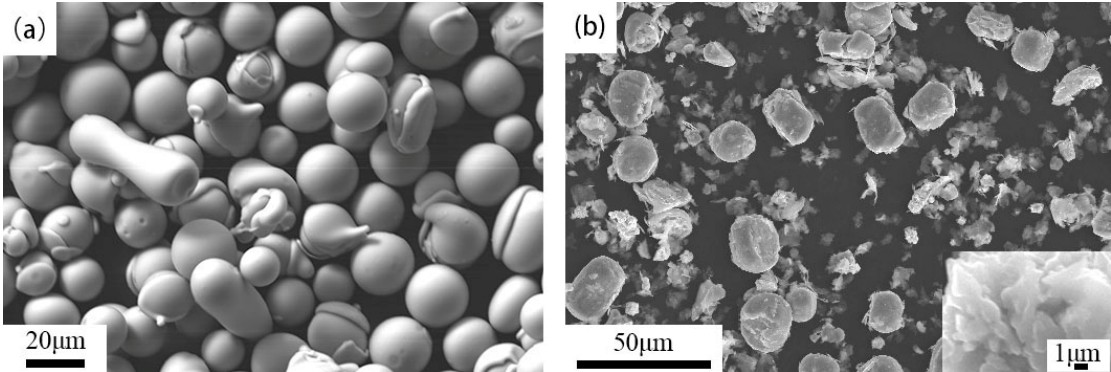

**Figure 2.** SEM image of (**a**) $Fe_{87}Si_5B_2P_3Nb_2Cu_1$ amorphous powder (**b**) $(Fe_{87}Si_5B_2P_3Nb_2Cu_1)_{86}Ni_{14}$ amorphous powder ball milled for 25 h.

Figure 3 is the EDS image of the surface of $(Fe_{87}Si_5B_2P_3Nb_2Cu_1)_{86}Ni_{14}$ amorphous powder. Figure 3a is the scan area image. Figure 3b is the Ni element distribution diagram. It can be seen from Figure 3 that the nickel element is uniformly distributed on the surface and around the amorphous powder. This result shows that the Ni micron powder and $Fe_{87}Si_5B_2P_3Nb_2Cu_1$ amorphous powder was completely fused with the increase of ball milling time.

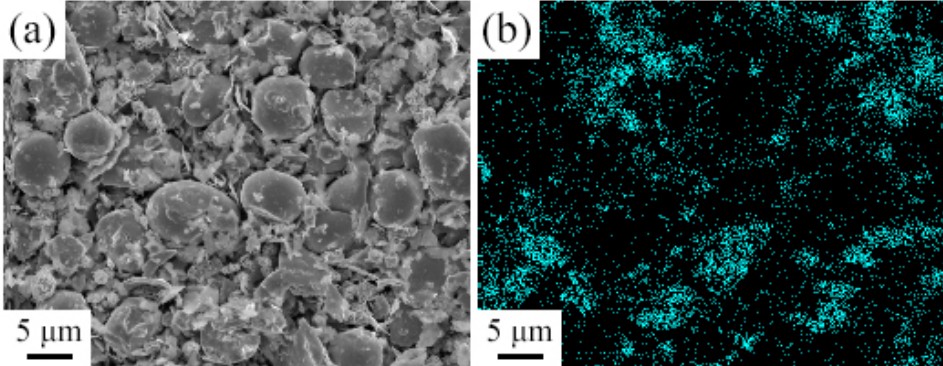

**Figure 3.** EDS image of $(Fe_{87}Si_5B_2P_3Nb_2Cu_1)_{86}Ni_{14}$ amorphous powder surface (**a**) scanned area image (**b**) Ni element distribution map.

Figure 4 shows the degradation rate of MB dyes at different temperatures with the reaction time changes. It can be seen from the Figure 4 that the degradation rate tended to be gentle under the reaction conditions of 40 °C. At 60 °C, the degradation rate gradually increased as the reaction continued. When the reaction reached 360 min, the MB dyes were almost completely degraded. When the temperature reached 80 °C, the initial reaction rate was the fastest. The $(Fe_{87}Si_5B_2P_3Nb_2Cu_1)_{86}Ni_{14}$ amorphous powder had the highest degradation rate to MB dyes. We found that the MB dyes were almost completely degraded at 80 °C for 180 min, and the degradation rate was as high as 99.99%. This result shows that the time required for complete degradation of MB dyes decreased with increased reaction temperature. Therefore, it was found that the experimental temperature had a drastic influence on the experimental result, and amorphous powder catalyst under the condition of 80 °C had the best performance for the degradation of MB among all the tested temperatures.

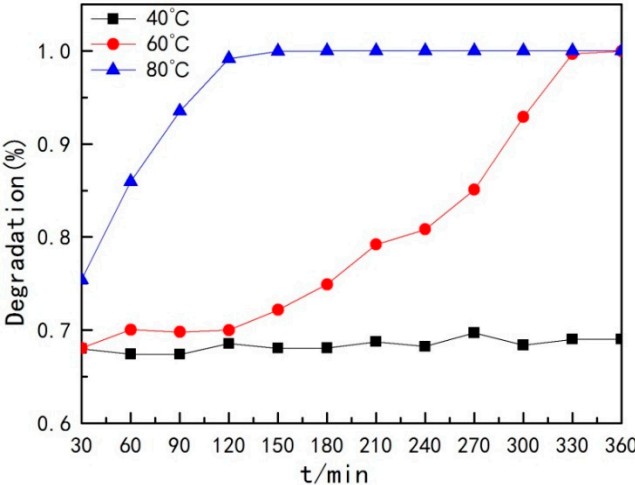

**Figure 4.** The degradation rate of Methylene blue (MB) dyes at different temperatures with the reaction time changes.

Figure 5 shows the UV-VIS absorption spectra of the amorphous powder catalyzed degradation of MB dyes. From Figure 5a, the reaction started very quickly because the amorphous powder had a smaller particle size after ball milling. Larger surface roughness and larger specific surface area promoted the reaction mixing with the MB solution. After 60 min of reaction, the concentration of the solution decreased rapidly and stabilized at 120 min. The inset of Figure 5a is a standard linear regression equation obtained by formulating different concentrations of MB solution. It can be seen from Figure 5b that the reaction was not obvious at the beginning. However, as the reaction proceeded, the reaction rate gradually increased, and the MB dyes were almost completely degraded at 330 min. It can be seen from Figure 5a,b that the $(Fe_{87}Si_5B_2P_3Nb_2Cu_1)_{86}Ni_{14}$ amorphous alloy could effectively degrade the MB dyes at 60 °C and 80 °C. With increasing reaction time, the intensity of the absorption peak at $\lambda_{max}$ = 664 nm gradually decreased to zero. The result showed that the concentration of MB dyes decreased, and the dye was completely degraded. However, the time required for the degradation of MB by the catalyst at 60 °C was double the degradation time required at 80 °C. This result demonstrates that temperature plays an important role in degradation experiments. From Figure 5c, the concentration of the MB solution did not change significantly in 240 min before the reaction, and then the reaction rate was accelerated to achieve dye degradation. Compared with Figure 5a,c, it was found that the degradation process of $Fe_{87}Si_5B_2P_3Nb_2Cu_1$ amorphous powder lagged far behind the degradation process of $(Fe_{87}Si_5B_2P_3Nb_2Cu_1)_{86}Ni_{14}$ amorphous powder under the same experimental conditions. Figure 5d is a UV-VIS absorption diagram of the degradation of MB dyes at 80 °C for $Fe_{87}Si_5B_2P_3Nb_2Cu_1$ amorphous powder ball milled for 25 h. The initial reaction rate was slow. When the reaction reached 270 min, the reaction rate suddenly increased, and MB dyes were rapidly degraded. Until 330 min, the absorbance was almost zero, and the reaction was completed. From Figure 5c,d, the sudden drop in dye absorbance had a long-term induction period in the degradation process. The reason for this phenomenon may have been that the specific surface area of the amorphous catalyst increased, and the surface physical adsorption was enhanced. Compared with Figure 5a,c, it was found that the MB dyes could be degraded under the same conditions, while $(Fe_{87}Si_5B_2P_3Nb_2Cu_1)_{86}Ni_{14}$ amorphous powder could accelerate the reaction rate and decrease the reaction time. That is to say, the catalytic degradation performances of $Fe_{87}Si_5B_2P_3Nb_2Cu_1$ amorphous powder were less sensitive to temperature than that of $(Fe_{87}Si_5B_2P_3Nb_2Cu_1)_{86}Ni_{14}$ amorphous powder.

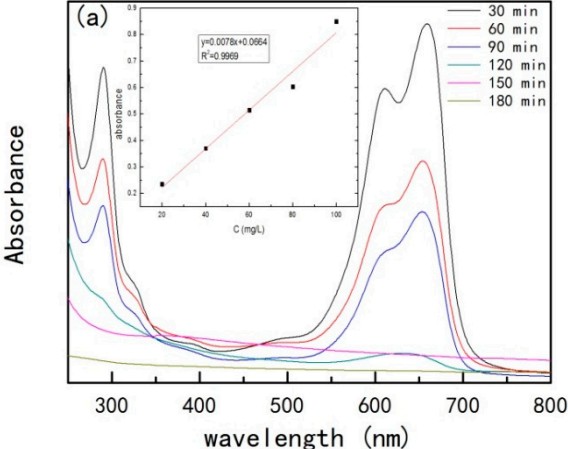

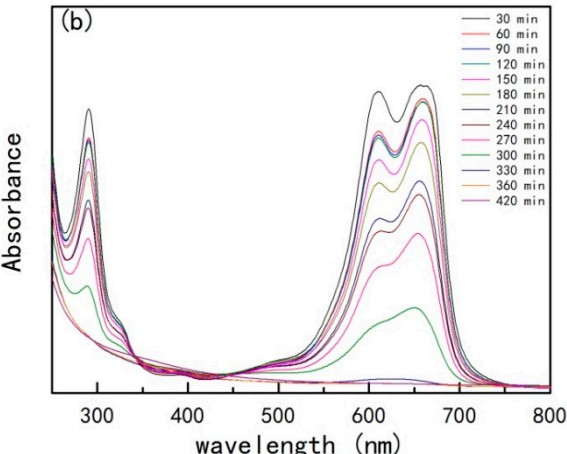

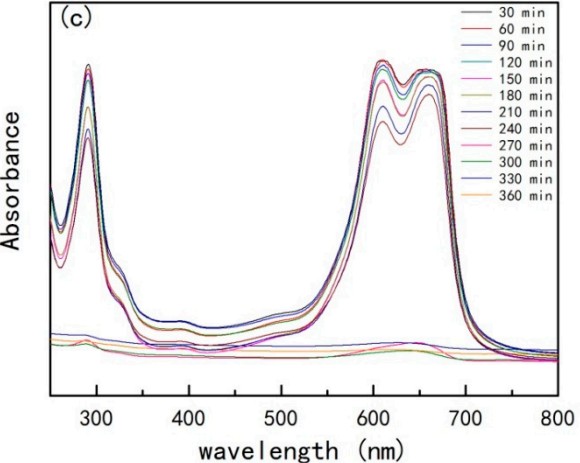

**Figure 5.** *Cont*.

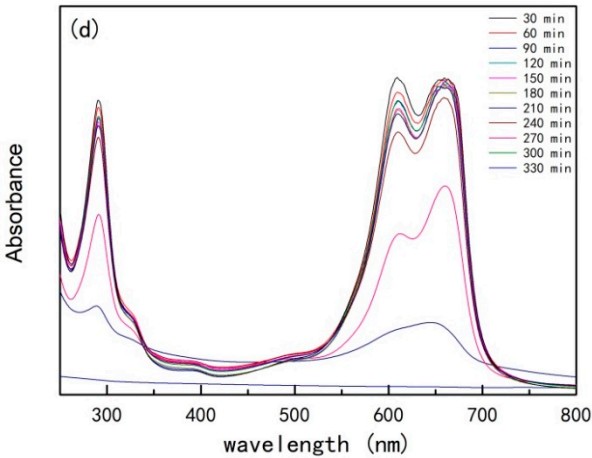

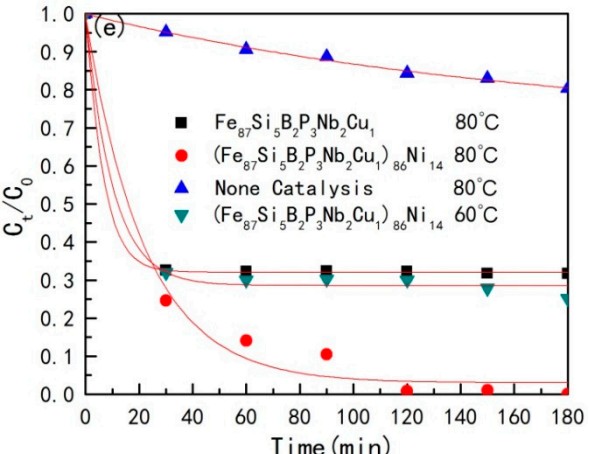

**Figure 5.** (**a**) UV-VIS spectra of MB with $(Fe_{87}Si_5B_2P_3Nb_2Cu_1)_{86}Ni_{14}$ amorphous powder at 80 °C (illustration is linear regression equation); (**b**) UV-VIS spectra of MB with $(Fe_{87}Si_5B_2P_3Nb_2Cu_1)_{86}Ni_{14}$ amorphous powder at 60 °C; (**c**) UV-VIS spectra of MB with $Fe_{87}Si_5B_2P_3Nb_2Cu_1$ amorphous powder at 80 °C; (**d**) UV-VIS spectra of MB with $Fe_{87}Si_5B_2P_3Nb_2Cu_1$ ball milled amorphous powder at 80 °C; (**e**) Kinetic curve ($C_t/C_0$).

Figure 6 is the UV-VIS spectrogram of the degradation of methylene blue dye by the recycled $(Fe_{87}Si_5B_2P_3Nb_2Cu_1)_{86}Ni_{14}$ amorphous alloy. It can be seen from Figure 6 that the initial rate of the reaction was too slow. Until the reaction proceeded to 90 min, the reaction rate suddenly increased, the absorbance decreased rapidly, and the dye was degraded. Until 210 min, the dye was barely degraded, and the reaction tended to be stable. The reaction lasted for 360 min, and the MB dyes were almost completely degraded. This may have been due to agglomeration of the surface of the recovered amorphous powder, a result of the relatively weak stability of the catalyst and the attenuation of the catalytic efficiency. Compared with Figure 5a, it was found that the recovered $(Fe_{87}Si_5B_2P_3Nb_2Cu_1)_{86}Ni_{14}$ amorphous alloy still had good catalytic activity. The degradation rate of MB dyes could reach 72.61%, and the amorphous alloy catalyst could be reused. However, the reaction rate was 0.015, which was only 1/4 of the catalytic rate of the $(Fe_{87}Si_5B_2P_3Nb_2Cu_1)_{86}Ni_{14}$ amorphous alloy. The recovered $(Fe_{87}Si_5B_2P_3Nb_2Cu_1)_{86}Ni_{14}$ amorphous alloy catalyst required longer reaction time to degrade MB dyes.

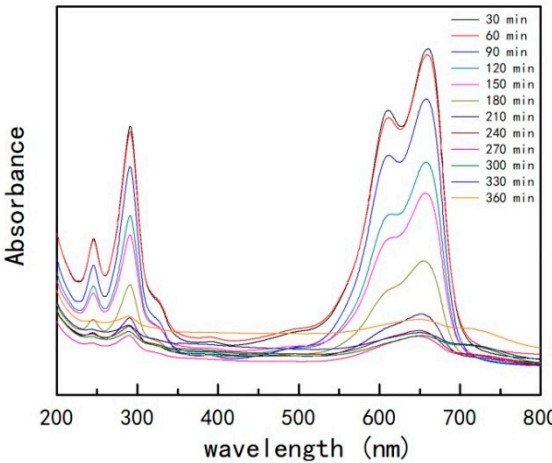

**Figure 6.** UV-VIS spectra of degradation of MB dyes by recycled $(Fe_{87}Si_5B_2P_3Nb_2Cu_1)_{86}Ni_{14}$ amorphous powder.

Figure 7 is the EDS image of the recovered $(Fe_{87}Si_5B_2P_3Nb_2Cu_1)_{86}Ni_{14}$ amorphous powder surface. Figure 7a is the scan area image. Figure 7b is the Ni element distribution diagram. A comprehensive analysis of Figures 3b and 7b shows that the distribution of the Ni element was not uniform, and the content was significantly reduced. This phenomenon indicates that Ni was corroded in the reaction and participated in degradation reactions, which accelerated the degradation rate of the MB dyes.

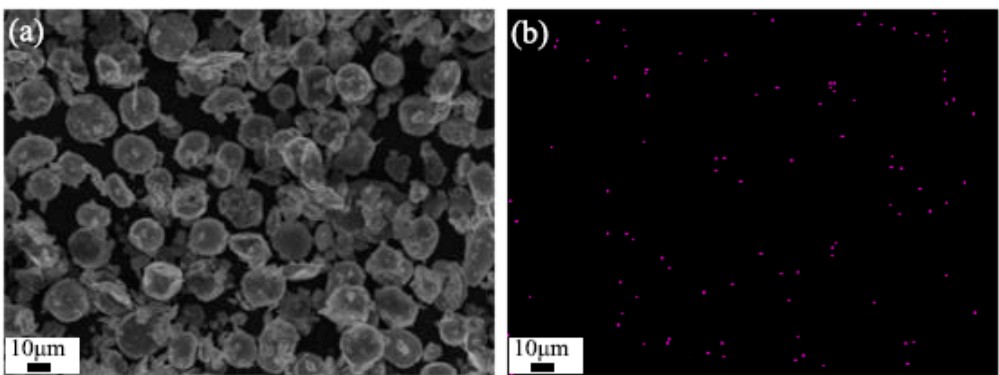

**Figure 7.** EDS image of the recovered $(Fe_{87}Si_5B_2P_3Nb_2Cu_1)_{86}Ni_{14}$ amorphous powder surface (**a**) scanned area image (**b**) Ni element distribution map.

*3.2. Formation of Mathematical Components*

The strong absorption peak at the $\lambda_{max} = 664$ nm of the visible light region of MB was derived from the conjugated structure formed by the azo bond, and the intensity therefore indicates the dye concentration in the solution. It can be seen from Figure 5 that the absorption intensity of the UV-VIS spectrum decreased as the reaction time increased during the degradation of the MB solution. The decreasing of peak strength at 664 nm implies the decreasing of dye concentration in the solution. No new absorption peaks appeared during the whole process, which suggests that no new degradation products entered the solution. The reaction mechanism is:

$$Fe^{2+} + H_2O_2 \rightarrow Fe^{3+} + \cdot OH + OH^- \tag{1}$$

$$Fe^{3+} + H_2O_2 \rightarrow Fe\text{-}O_2H^{2+} + H^+ \tag{2}$$

$$Fe\text{-}O_2H^{2+} \rightarrow Fe^{2+} + \cdot O_2H \tag{3}$$

In Fe-Si-B amorphous alloys, $Fe^{2+}$ catalyzes the decomposition of $H_2O_2$ to generate the •OH radical, and $Fe^{2+}$ oxidizes to $Fe^{3+}$ ion after the reaction formula (1). The latter reacts with $H_2O_2$ in the aqueous solution to form the intermediate $Fe-O_2H^{2+}$. The intermediate decomposes into $Fe^{2+}$ and •$O_2H$ radicals under agitation of the rotor. This is similar to the Fenton reaction, which increases the •OH radical in the reaction system. The addition of $Fe^{2+}$ significantly improves the decolorization and degradation efficiency of the dye, and the degradation of MB dyes is dominated by the oxidation of the •OH radical.

Figure 5e shows the kinetics curve of MB solution degradation. Wherein $C_0$ is the initial concentration of the MB solution, and $C_t$ is the concentration of MB remaining in the solution at the reaction time of $t$. It was found by nonlinear curve fitting that the residual rate of MB concentration changed with time according to the first order reaction model in chemical reaction kinetics. The relevant expressions are as follows [24]:

$$C_t/C_0 = \exp{(-k \cdot t)} \tag{4}$$

$$t_{1/2} = ln2/k \tag{5}$$

$$\text{Degradation rate} = (C_0 - C_t)/C_0 \tag{6}$$

where $k$ is the reaction rate constant, $t$ is the reaction time, and $t_{1/2}$ is half-period. The relevant kinetic parameters $k$, $t_{1/2}$, and degradation rate obtained by fitting Figure 5e are listed in Table 1. As can be seen from Table 1, the reaction rate of $(Fe_{87}Si_5B_2P_3Nb_2Cu_1)_{86}Ni_{14}$ amorphous powder on MB dyes was 3.6 times larger than the $Fe_{87}Si_5B_2P_3Nb_2Cu_1$ amorphous powder and 65 times larger than the non catalyst. Its chemical reaction half-period was 10.66 min, which was close to 1/3 of the half-period of the $Fe_{87}Si_5B_2P_3Nb_2Cu_1$ amorphous powder. The reaction rate of $(Fe_{87}Si_5B_2P_3Nb_2Cu_1)_{86}Ni_{14}$ amorphous powder to MB dyes at 80 °C was 2.9 times larger than that of the $(Fe_{87}Si_5B_2P_3Nb_2Cu_1)_{86}Ni_{14}$ amorphous powder at 60 °C, which was also close to 1/3 of the half-period of it.

**Table 1.** Different degradation reaction kinetic parameters.

| Material | Temperature (°C) | $k$ | $t_{1/2}$ (min) | Degradation Rate (%) |
|---|---|---|---|---|
| None catalysis | 80 | 0.001 | 693.15 | 19.68 |
| $Fe_{87}Si_5B_2P_3Nb_2Cu_1$ | 80 | 0.018 | 38.51 | 67.76 |
| $(Fe_{87}Si_5B_2P_3Nb_2Cu_1)_{86}Ni_{14}$ | 80 | 0.065 | 10.66 | 99.99 |
| $(Fe_{87}Si_5B_2P_3Nb_2Cu_1)_{86}Ni_{14}$ | 60 | 0.022 | 31.51 | 99.95 |

Analysis of Figure 5e shows that when other experimental conditions were unchanged, the MB dyes gradually degraded under the action of $H_2O_2$ oxidant and rotor stirring without adding any catalyst. However, the reaction proceeded very slowly, and the degradation rate was only 19.68% after 180 min. The degradation rate of MB dyes was obviously improved by adding amorphous samples. After the reaction to 180 min, the degradation rate of MB dyes by $Fe_{87}Si_5B_2P_3Nb_2Cu_1$ amorphous powder reached 67.76%, and the degradation rate of MB dyes by $(Fe_{87}Si_5B_2P_3Nb_2Cu_1)_{86}Ni_{14}$ amorphous powder was as high as 99.99%. It can be seen that the addition of a catalyst could significantly improve the degradation efficiency. However, the $Fe_{87}Si_5B_2P_3Nb_2Cu_1$ and $(Fe_{87}Si_5B_2P_3Nb_2Cu_1)_{86}Ni_{14}$ amorphous alloy catalysts had rapid decline and then stable states in the process of degrading the dye. The comprehensive reaction mechanism analysis shows that the reasons for this phenomenon might have been attributed to the presence of Ni, which may have promoted the oxidation of $Fe^{2+}$ to $Fe^{3+}$ and the generation of a large number of •OH radicals. Ni rapidly participated in the chemical reaction at the initial stage of the reaction, which was not conducive to the reduction of $Fe^{3+}$ to $Fe^{2+}$. However, when the temperature increased, it made up the difference—exactly—between the reduction process and the oxidation process.

At the same time, comparing the $Fe_{87}Si_5B_2P_3Nb_2Cu_1$ amorphous alloy, the catalytic activity and catalytic efficiency of $(Fe_{87}Si_5B_2P_3Nb_2Cu_1)_{86}Ni_{14}$ amorphous alloy after ball milling for 25 h were improved. The degradation rate increased from 67.76% to 99.99%, which was almost completely

degraded. Further work will be done to explain the mechanism of chemical degradation on azo dye solution from the perspective of electronic structure effects.

## 4. Conclusions

In summary, we prepared $(Fe_{87}Si_5B_2P_3Nb_2Cu_1)_{86}Ni_{14}$ amorphous alloy by ball milling in a certain ratio of $Fe_{87}Si_5B_2P_3Nb_2Cu_1$ and high-purity Ni micron powder. We studied the influence on the catalytic performance of the amorphous alloy for degraded MB dyes. It was proven that the $(Fe_{87}Si_5B_2P_3Nb_2Cu_1)_{86}Ni_{14}$ amorphous powder possessed higher roughness and ripple. The specific surface area of the amorphous alloy increased. At the same time, the contact area between the catalyst and the solution increased, which further promoted the diffusion of the reaction and the generation of more active sites for the degradation reaction. All these effects increased the catalytic activity of amorphous powder.

By a series of comparative experiments, it was found that the optimum temperature for degrading the amorphous catalyst to 100 mg/L MB dyes was 80 °C. All of the amorphous powder could degrade the MB dyes under these conditions. However, the reaction rate and degradation rate were different, which led to different reaction times as well. Reaction kinetics of catalytic degradation of MB dyes by amorphous powders were in accordance with the first-order reaction model. The reaction rate of $(Fe_{87}Si_5B_2P_3Nb_2Cu_1)_{86}Ni_{14}$ amorphous powder to MB dyes was three times larger than thatof the $Fe_{87}Si_5B_2P_3Nb_2Cu_1$ amorphous powder. The half-period of the chemical reaction was only one third of the $Fe_{87}Si_5B_2P_3Nb_2Cu_1$ amorphous powder. The degradation rate increased from 67.76% to 99.99%, which achieved dye degradation. These finding have important application prospects for further improving the catalytic performance of amorphous catalysts in the treatment of industrial wastewater.

**Author Contributions:** Methodology, B.N.; formal analysis, J.Z.; data curation, Y.C.; writing—original draft preparation, J.S.; writing—review and editing, Z.Z.; supervision, C.W. and D.C.; funding acquisition, H.W. and M.W.

**Funding:** This research was funded by the National Natural Science Foundation of China (51671035) and the National Natural Science Foundation of China (51601018).

**Acknowledgments:** In this section, I can acknowledge any support given by Jilin Provincial Government and the support of every teacher to this research project.

**Conflicts of Interest:** The authors declare no conflict of interest

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
