# Peer review of "Effect of Ni Addition on Catalytic Performance of Fe87Si5B2P3Nb2Cu1 Amorphous Alloys for Degrading Methylene Blue Dyes"

_metals, doi:10.3390/met9030341_

Round 1

Reviewer 1 Report

This revised version of your manuscript was greatly improved, but you have still a lot of careless mistakes.

at line 35: "people's health" should be "human health"

at line 92: "and" should be "and"

at line 102: "0.31mol/L" should be "0.31 mol/L"

at line 120: 3.1 "Subsection" should be changed correctly

at line 194: "significantly 240 min" may be "significantly in 240 min"

at line 231 in Table: t1/2 has a unit of min, so t1/2 (min), kobs, t1/2 should be kobs, t1/2

at line 246: "02H radicals" may be "●02H radicals"

at line 250: "C0" should be "C0"; All the physical quantities should be written in an italic style both in text and equation. Many corrections are required at lines 254-258.

Author Response

Based on the reviewer's good advice, we have modified the writing of all physical quantities in the manuscript, and the modified parts are marked in red in the re-uploaded manuscript. 

And other inappropriate places in the article have also been corrected (For example: line spacing and other formats). 

You can find our point-by-point reply to the reviewer's comments in the word document below.

Reviewer 2 Report

The concept of the manuscript by Zhao and coworkers is not a bad one, and certainly fitting within the context of the journal metals. The synthesis of the materials is well done, however, pertinent information detailing the ball-milling step are missing. Such as free volume of reactor, grinding media, etc... These are all critical to the reproducibility of such work, and are currently lacking in the manuscript. Additionally, Figure 1 is difficult to interpret. It would benefit from a second inset which has a "zoomed-in" view of the Ni(111), Fe(110) and Ni(200) peaks.

Finally the interpretation of the catalytic data is unclear. It is not clear based on Figures 5b-d what values the authors used to extract catalytic rates in order to compare their materials. Also, according to Figure 5c and d, it appears those materials are actually very catalytically active (moreso than the Ni containing alloys), but with a very long induction period. Finally, no efforts were made to address recyclability which is critical for the application of any solid catalyst. Such studies would yield valuable information. For instance, it may tell us if the activity is potentially due to leached nickel particles (which may degrade azo and nitro functionalities). For the moment, it is unclear what the actual effect of the Ni species is, only that there is one.

It is my recommendation that prior to publication, these issues must be addressed. Specifically, the degradation rates should be plotted clearly (Figure 5 e is incomplete, and misleading). Also the authors should endeavor to explain why they are observing a sudden drop in absorbance which plateaus into a long induction period.... is this simply rapid initial surface adsorption? How do the authors reconcile the fact that with no catalyst, there is steady degradation, yet in the presence of catalyst, there is a fast drop then plateau?

Author Response

Based on the reviewer's good advice, we supplemented the experiments and tests related to dye degradation by recycled amorphous catalysts. 

And other inappropriate places in the article have also been corrected (we convert all the data graphs in the manuscript into JPG format, and according to the template, the line spacing and other formats are modified accordingly, etc.).

At the same time, the supplemental data and the modified part are marked in red in the re-uploaded manuscript. 

You can find our point-by-point responses to reviewers' comments in the word document below.

Round 2

Reviewer 2 Report

The changes by the authors address the previous comments. There are some grammatical English language issues which should be carefully addressed before the final version goes online. Otherwise, I can recommend publication in Metals in it's current form after minor grammatical revision.

Author Response

According to the good guidance of reviewers, we have revised the questions about English in the newly uploaded manuscript.

It mainly includes tenses, singular and plural nouns and so on.

The inaccuracies described in the manuscript also were corrected.

You can find our point-by-point responses to reviewers' comments in the word document below.

This manuscript is a resubmission of an earlier submission. The following is a list of the peer review reports and author responses from that submission.

Round 1

Reviewer 1 Report

This manuscript describes to synthesis (Fe87Si5B2P3Nb2Cu1)86Ni14 by ball milling method and to investigate a catalytic performance for degrading Methylene Blue Dyes. This amorphous alloy is compared to degrade MB Dyes with Fe87Si5B2P3Nb2Cu1, and the effect of Ni addition to Fe87Si5B2P3Nb2Cu1 is valid on improving to degrade MB Dyes. The technique is an available result, and the authors express that (Fe87Si5B2P3Nb2Cu1)86Ni14 can obtain optimal catalytic degradation properties. However, the authors’ original scope for the optimization of the catalyst added Ni to Fe87Si5B2P3Nb2Cu1 in this study should be mainly clarified about the scientific locations of the effort among the authors’ results. For above reasons, this paper needs some revisions for the publication standard of this journal at the moment.

1.       Please enlarge Figures. 1, 3 and 4 in the manuscript. Especially, please indicate the peak positions of alfa-Fe and Ni of XRD clearly.

2.       The authors should express the composition distribution of Ni in Fe87Si5B2P3Nb2Cu1 in Figure 2. Ni is contained into particles of approximately 10~20 mm in Fig.2(b)?  

3.       The inset in Fig.4(a) should be clearly. The reviewer could not find it.

4.       The authors should explain scientifically the reason of optimizations of the catalytic performance Discussion or Conclusions chapter of your manuscript.  

Author Response

Response to Reviewer Comments

Manuscript ID: metals-438076

Title: Effect of Ni Addition on Catalytic Performance of Fe87Si5B2P3Nb2Cu1 Amorphous Alloys for Degrading Methylene Blue Dyes

Dear Editor and Reviewers,

Thank you for your letter and for the reviewer’s comments concerning our manuscript. Those comments are all valuable and very helpful for revising and improving our paper. We have studied comments carefully and have made corrections accordingly. And other inappropriate places in the article have also been corrected. These revised parts are marked in yellow in the paper. You will find our point-by-point responses to the reviewer’s comments in the word below:

Reviewer 2 Report

The paper describes the effect of Ni addition on catalytic properties of Fe87Si5B2P3Nb2Cu amorphous alloys for decomposition of methylene blue. The Ni addition effect can be clearly seen in Figure 4. However, discussion on the origin of the effect is not enough. The author mentioned the effect, rapid degradation, was obtained by roughened surface, which did not result from Ni addition, but from mechanical ball milling. To show the originality and validity of this paper, the reaction rate and surface roughness as well as XRD pattern of the sample should be presented as a function of ball milling time. In addition, some issues listed below must be considered before acceptance for publication.

1.  In Figure 1, XRD patterns are too small to judge your arguments. In particular, readers cannot obtain line widths from Figure 1. The line located at 53 degrees corresponds to diffraction of Ni (200).   

2. In Figure 3, two graphs of degradation rate versus time for 60 and 80 degrees are completely different. In Figure 4 (d), the data for 80 degrees are fitted by the equation (4) that obeys first order kinetics. Is it possible to fit the data for 60 degrees in the same way? In Table 1, both kobs and t1/2 for 60 degrees were obtained. The reviewer does not know the reason why the data for 60 degrees are missing in Figure 4 (d).

3. The manuscript is incomplete. There are many careless mistakes. For example, in title (Line 3), Fe87Si5B2P3Nb2Cu1 must be Fe87Si5B2P3Nb2Cu. In Lines 110 and 111, “Subsection” and “Subsubsection” should be corrected. In my opinion, the section “4. Discussion” can be omitted because contents are not enough.

Author Response

Response to Reviewer Comments

Manuscript ID: metals-438076

Title: Effect of Ni Addition on Catalytic Performance of Fe87Si5B2P3Nb2Cu1 Amorphous Alloys for Degrading Methylene Blue Dyes

Dear Editor and Reviewers,

Thank you for your letter and for the reviewer’s comments concerning our manuscript. Those comments are all valuable and very helpful for revising and improving our paper. We have studied comments carefully and have made corrections accordingly. And other inappropriate places in the article have also been corrected. These revised parts are marked in yellow in the paper. You will find our point-by-point responses to the reviewer’s comments intheword document below:

Reviewer 3 Report

This article is not suitable for publication in its present state.

It has sections that look like they're just drafts with missing titles, incomplete sentences, inconsistent bulleted lists etc. Moreover, the article is written in clumsy English and parts that are well written contain obvious plagiarism (one article that is blatantly copied in the introduction isJournal of Materials Science & Technology, 2017, 33(8): 856-863). I did not run a thourough check for plagiarism (I assume the editors will do it... ?) but several sentences I searched were pure and simple copy/paste efforts.

All this does not indicate anything good about the article or its authors. 

I think the paper should be fully reviewed, completed as any proper article should be and cleansed of all plagiarism before publication can even be considered.

Author Response

Response to Reviewer Comments

Manuscript ID: metals-438076

Title: Effect of Ni Addition on Catalytic Performance of Fe87Si5B2P3Nb2Cu1 Amorphous Alloys for Degrading Methylene Blue Dyes

Dear Editor and Reviewers,

Thank you for your letter and for the reviewer’s comments concerning our manuscript. Those comments are all valuable and very helpful for revising and improving our paper. We have studied comments carefully and have made corrections accordingly. And other inappropriate places in the article have also been corrected. These revised parts are marked in yellow in the paper. You will find our point-by-point responses to the reviewer’s comments in the word document below:

Round 2

Reviewer 2 Report

The revised manuscript has been satisfactorily improved. The reviewer needs no more request to authors.The paper would be interesting for the readers of the Metals.

Reviewer 3 Report

The paper has been improved but there are still multiple mistakes (missing titles etc.) that reflect poor care in the preparation of the MS. This is no proper way to prepare and submit a MS. 

I urge the editor to run a real plagiarism detection test on the article. That's not my job to do, but as it stands I don't want to support publication of an article with obvious plagiarism. For example, just google the sentence "Because of the growing environmental concerns and increasing public pressure, the removal of all unsightly pollutants from their effluent outfalls is a prime concern to all industrial and manufacturing organizations." taken from the introduction and you'll see it was copied (plain and simple) from a previous article that I pointed out in my first report. This is unacceptable. Period.

If the editors of Metals want to support this kind of publication, that's their responsibility. I don't agree.